# Wage Premiums as a Means to Evaluate the Labor Market for Pharmacy Technicians in the United States: 1997–2018

**DOI:** 10.3390/pharmacy8010042

**Published:** 2020-03-17

**Authors:** David P. Zgarrick, Tatiana Bujnoch, Shane P. Desselle

**Affiliations:** 1Department of Pharmacy and Health Systems Sciences, School of Pharmacy, Bouvé College of Health Sciences, Northeastern University, Boston, MA 02115, USA; 2Division of Pharmacy, Memorial Hermann Health System, Houston, TX 77024, USA; Tatiana.bujnoch@memorialhermann.org; 3Department of Social, Behavioral & Administrative Sciences, College of Pharmacy, Touro University California, Vallejo, CA 94592, USA; shane.desselle@tu.edu

**Keywords:** pharmacy technicians, labor market, wages

## Abstract

Pharmacy technicians are integral members of the health care team, assisting pharmacists and other health professionals in assuring safe and effective medication use. To date, evaluation of the labor market for pharmacy technicians has been limited, and relatively little has been evaluated regarding trends in wages. The objective of this research is to use US Bureau of Labor Statistics (US BLS) data to evaluate changes in pharmacy technician wages in the United States from 1997 to 2018 relative to changes in the US consumer price index (CPI). Median hourly wages for pharmacy technicians were collected from US BLS data from 1997 to 2018. Median hourly wages were compared to expected hourly wages, with the difference, a wage premium, indicative of imbalances in the supply and demand of labor. Both positive and negative wage premiums were observed, with most positive wage premiums occurring prior to 2007 and most negative wage premiums observed after 2008. Differences in wage premiums were also observed between technicians working in various practice settings. Given the median length of employment of pharmacy technicians, it is likely that the majority of technicians working in US pharmacies have not experienced increases in their wages relative to what would be expected by changes in the CPI. This has occurred at a time when pharmacies and pharmacists are asking more of their pharmacy technicians. Researchers and pharmacy managers must continue to evaluate the pharmacy technician labor market to assure that technician wage and compensation levels attract an adequate supply of sufficiently skilled workers.

## 1. Introduction

The labor markets for health care workers are important to monitor and evaluate, as these workers are still the driving force behind the delivery of goods and services that improve the health of individuals and entire populations. This is particularly important in the profession of pharmacy, where the increased reliance on medications as a form of treatment requires trained personnel to ensure that medications deliver desired outcomes and avoid undesired outcomes.

Some degree of attention has been paid over the years to the labor markets for health care professionals, such as those for physicians, pharmacists, nurses and others who are educated and trained to play specific roles in serving our health needs [1,2,3]. As health care has become more specialized and complex, health professionals have increasingly relied on para-professional workers to support their clinical roles and provide administrative assistance. In pharmacy, the role of the pharmacist has been increasingly supported by the pharmacy technician. Pharmacy technicians support pharmacists in dispensing medications, performing clinical functions needed to improve the outcomes of mediation use [4,5,6], and to perform a number of administrative functions which assist in the operations of a pharmacy [7]. Pharmacy technicians have been asked to increasingly take on roles that had previously been exclusively performed by pharmacists, including reviewing medications and checking the work of other technicians prior to dispensing [8,9], taking medication histories [10], managing warfarin therapy within a clinical pharmacy anticoagulation service [11] and immunization delivery [12]. Pharmacists depend upon a stable labor market of pharmacy technicians to support them in optimizing patient health outcomes. The ability to delegate and empower others is demonstrative of a pharmacist practicing at the top of their license [13].

The number of pharmacy technicians working in the United States has grown from 165,400 in 1997 to 420,400 in 2018 [14]. The number of pharmacy technicians in the US now surpasses the number of pharmacists (314,300). The United States Bureau of Labor Statistics (US BLS) projects that the job market for pharmacy technicians will grow by 7% between 2018–2028, adding 31,500 positions [15]. At the same time that the US BLS is not projecting any net growth in the number of pharmacist positions needed in the US [16].

The supply and demand of health care professionals has been the subject of considerable research. The labor market for pharmacists in the United States has been evaluated by means of the aggregate demand index (ADI) [17], and later by the pharmacist demand indicator (PDI) [18]. Pharmacist demographics, working conditions and other trends in pharmacist practice in the US are examined in the National Pharmacist Workforce Survey, which has been conducted every five years since 2000 [19]. Our understanding of the supply and demand of pharmacy technicians in the US and their working conditions are more limited. Desselle and Holmes conducted a National Certified Pharmacy Technician Workforce Survey in 2015 which described various aspects of their working conditions, including a finding that over one in four certified pharmacy technicians (26.6%) were “highly dissatisfied” with their wages [20]. Urick and colleagues noted that while many US states adopted additional barriers to entry to working as a pharmacy technician between 1997 and 2017, such as registration with a state entity and national certification requirements, these barriers were not associated with any changes in pharmacy technician wages [21]. Mattingly and Mattingly also found that that were no significant differences in pharmacy technician wages in 2016 based on the degree of regulation of pharmacy technician practice in that state or the cost of living in a state as measured by the salary housing index [22]. Mattingly and Mattingly also concluded in their systematic review of literature regarding the roles of pharmacy technicians that while evidence supports technicians performing roles which advance pharmacy practice and improve patient outcomes, the benefits to technicians in performing these roles have been limited to increases in their job satisfaction and work schedules, and not in their levels of wages or other forms of financial compensation [23]. They further concluded that if pharmacy technicians are to take on more roles in the future, they may need to be offered more tangible forms of benefits, particularly if these roles require completion of formalized education and training programs.

Limitations of much of the previous research on pharmacy technician labor is that it is cross sectional and only describes labor market conditions at a particular point in time. Little research has been done to evaluate how pharmacy technician wages have changed over time, and how that in turn this has been reflected by changes in the number of technicians leaving or entering the labor market. Even more scarce is research evaluating trends in various sectors of the labor market for pharmacy technicians, sectors which can be defined by the setting in which the work takes place (e.g., chain and independent pharmacies, grocery store pharmacies, mass merchandise store pharmacies, hospitals, government agencies).

Since 1997, the US BLS has collected data annually on over 800 occupational groups, including pharmacists and pharmacy technicians. Among the data collected by the BLS for each occupational group are the mean and median annual salary and hourly wage levels, as well as the number employed in that group. US BLS occupational group data can be further analyzed by workplace setting. The objective of this research is to use US BLS data to evaluate changes in pharmacy technician wages in the United States from 1997 to 2018 relative to changes in the US consumer price index (CPI) over that time. The underlying hypothesis for this comparison is that if the supply and demand for labor are in balance, changes in wages for that occupation will match changes in the CPI. If differences between an occupation’s wages from what would be predicted by changes in CPI are noted, that would be a signal that the supply and demand are not in balance. For example, increases in real wages over those predicted by CPI could be explained by both shortages of labor, and/or increased demand for that service. Decreases in real wages relative to CPI may be indicative of just the opposite; a combination of oversupplies of labor in that market and/or decreased demand for that particular type of labor.

## 2. Methods

Each May since 1998 the US BLS has released occupational employment statistics (OES) for over 800 occupational groups, reflecting information collected in May of the preceding year [24]. From this OES data the median hourly wage for pharmacy technicians (US BLS OES code 29-2052) was collected for each year from May 1997 through May 2018 [25]. In addition to collecting median hourly wages for all pharmacy technicians, median hourly wages were also collected each year for pharmacy technicians employed in industry sectors, including chain and independent pharmacies, grocery store pharmacies, mass merchandise pharmacy, hospitals, and government agencies. The percentage change in the US consumer price index (CPI) from May to May of each year from 1997 through 2018 was also gathered from US BLS [26].

Beginning with 1997, the actual median pharmacy technician hourly wage was multiplied by the percentage change in CPI over the next year to determine the expected median pharmacy technician hourly wage for the following year. This expected median hourly wage was then multiplied by the CPI for each subsequent year to determine the median hourly wage that would have been expected for each year from the base year through May 2018. The actual median pharmacy technician hourly wage for each following year was then compared to the expected median pharmacy technician hourly wage for that year as calculated above. This is demonstrated in Figure 1, in which, beginning with the actual median pharmacy technician hourly wage in 1997, expected median hourly wages for future years were calculated, and then comparisons were made to the actual median hourly wage pharmacy technicians had each year. The primary research hypothesis is that when these comparisons are made the difference will be $0, reflecting a balance in compensation paid to these employees and their ability and willingness to accept these wages in the market. Any differences found between the actual median hourly wage and the expected median hourly wage for any particular year is defined as a wage premium (wage premium = actual median hourly wage – expected median hourly wage). Wage premiums which occurred prior to 2018 were adjusted to reflect 2018 net present values. The adjusted wage premiums were then summed from the base year through 2018, and then divided by the number of years being analyzed to determine the mean wage premium experienced by pharmacy technicians for each year over the term being considered. In the example described in Figure 1, pharmacy technicians in the United States experienced a mean adjusted wage premium of $1.99/hour between 1997 and 2018, meaning that the median hourly wage for pharmacy technicians who worked between these years was, on average, $1.99/hour higher than would had been expected had their wages increased by the CPI each year.

Figure 2 describes changes in the median hourly wages from 2009 to 2018. Pharmacy technician median hourly wages in the United States decreased by an average of $0.35/hour relative to what would have been expected had their median hourly wage increased by the CPI over that time.

This analysis was repeated for all base years from 1997 to 2017, resulting in ranges from one year (2017 to 2018) to twenty-one years (1997 to 2018). This analysis was also performed on subsets of data for pharmacy technicians working in chain and independent pharmacies, grocery store pharmacies, mass merchandise store pharmacies, hospitals, and for government agencies. An analysis was also performed to evaluate the presence of wage premiums across all workers in all occupational groups in the United States.

## 3. Results

Figure 3 represents trends in the median hourly wages received by pharmacy technicians between 1997 and 2018. Median hourly wages across all pharmacy technicians have increased from $8.39/hour in 1997 (which, adjusting based on changes in the US CPI is the equivalent to $13.18/hour in 2018) to $15.72/hour in 2018. Variation in median hourly wages is noted between the settings where pharmacy technicians work. Median hourly wages for pharmacy technicians in 2018 ranged from $14.65/hour in chain and independent pharmacies and $14.73/hour in food store pharmacies to $17.97/hour in hospitals and $21.10/hour in government settings. It should also be noted that the median hourly wage in the United States in May 2018 for all workers across all occupations was $18.58/hour [24].

Figure 4 represents the trends in wage premiums in pharmacy technician hourly wages which occurred between 1997 and 2018. Pharmacy technician median hourly wages consistently experienced positive wage premiums from base years prior to 2007, meaning that the median hourly earnings of technicians from the base year to 2018 were higher than would had been expected had their hourly wages increased by the CPI. Since 2007, pharmacy technicians in the United States have been much less likely to experience positive wage premiums. In the 2009 and 2010 base years, median hourly wages for pharmacy technicians were $0.35/hour and $0.25/hour less than what would had been expected than if their wages had kept up with the CPI over that time. The wage premiums experienced by pharmacy technicians since 2006 have been very similar to those experienced by all workers in all occupations in the US over that time.

The presence and absence of wage premiums also varied between sectors of the US pharmacy technician workforce. Pharmacy technicians in grocery store settings have been experiencing negative wage premiums since 2000, some earning over $1.00/hour less than would have been expected given changes in CPI. On the other hand, pharmacy technicians in mass merchandiser settings experienced higher wages than would have been expected by changes in CPI for all years up to 2016.

## 4. Discussion

While pharmacy technicians in the United States who began their work between 1997 and 2006 have likely experienced modest positive wage premiums over time, pharmacy technicians who begun their work after 2007 have likely not experienced any positive wage premiums, with some earning less than what would have been expected had their wages kept up with the CPI. It should be noted that pharmacy technicians who have been working in this role since the late 1990s and early 2000s likely account for a very small portion of the pharmacy technician workforce in the United States. According to data from Desselle and colleagues, the median length of employment as a pharmacy technician in the US is approximately 9–10 years [27] compared to a median length of employment of approximately 22 years for pharmacists [28]. The majority of pharmacy technicians in the US have entered the workforce since 2006, and as such have not experienced much in the way of positive wage premiums, and may have even experienced negative wage premiums. Pharmacy technicians who have entered the work force since 2006 may have benefitted indirectly from hourly wage levels that were higher in CPI-adjusted dollars than if had they started in the field earlier. Since 2006 wage premiums for pharmacy technicians in the US have fluctuated in a manner and to a degree similar to those experienced by the median of “all occupations” in the US economy. Yet, what has been asked of pharmacy technicians in terms of expanded job roles [4,5,6,7,8,9,10,11,12] and regulatory requirements [21,22,23] has increased over this time. While one would expect to see positive wage premiums as a result of these increases in job roles and requirements, this was not observed in our analysis, and is consistent with the findings of other researchers [21,22]. Pharmacy technicians in the United States also have not experienced wage changes in the manner of those experienced by pharmacists, who experienced relatively large positive wage premiums in the late 1990s through the late 2000s [29].

Pharmacy technicians employed in community pharmacy settings in the United States (grocery stores, chain and independent pharmacies, mass merchandise stores) have generally experienced lower hourly wages and have been more likely to experience negative wage premiums than pharmacy technicians in hospitals and government settings. Community pharmacy technicians have been described as “the face of the community pharmacy”, which translates into them being a critical source of patient loyalty, satisfaction and engagement. Their expanded job scope has resulted in higher levels of job stress [30]. As pharmacy technicians are essential to community pharmacy in the midst of their practice becoming more patient-centric model, community pharmacies must compensate their technicians at levels in line with the responsibilities they are increasingly being asked to take on, or risk losing them to other practice settings (such as hospitals and government, which has consistently paid pharmacy technicians higher hourly wages) or to other jobs outside of pharmacy entirely. Mattingly and Boyle have found that pharmacy technicians were the lowest compensated of 14 health technologist occupations in the U.S. state of Maryland, despite having education, training, and regulatory requirements similar to those other occupations [31]. Chui and colleagues stated that community pharmacies must redesign technician jobs, deploy them more effectively, and provide the training and compensation commensurate with these jobs so as to re-engineer practice with greater patient safety in mind [32]. In a 2005 study, Desselle found that as little as a $0.75/hour difference in hourly wages (approximately $1.00/hour when converted to 2018 levels) explained a significant portion of a pharmacy technician’s intention to remain with their current employer or seek other employment [33].

This study has limitations that are important to recognize. The US Bureau of Labor Statistics Occupational Employment Statistics are exclusively collected in the United States and its territories. The results described here are limited to pharmacy technicians in the United States. The manner in which the US BLS provides its data (it is over one year old at the time it first becomes available), and the subsequent use of this data to calculate the presence of positive or negative wage premiums means that the findings reported here are a trailing indicator of the state of the labor market in the past. This study of US BLS data precludes any ability to discern from pharmacy hiring managers and other stakeholders the realities of the hiring market at the present time. While this analysis enables identification of past incongruities in the labor market (e.g., groups who have experienced either positive or negative wage premiums), the analysis does not allow us to separate to the extent to which wage premiums (or lack thereof) can be explained by changes in the supply of labor willing to work at these wage levels, or changes in the demand for services provided by pharmacy technicians. There is ample evidence that the demand for the services of pharmacy technicians in the United States has been high during this time period. The fact that significant positive wage premiums have not been observed, particularly since 2006, may be due a variety of factors. These include increases in the supply of pharmacy technician labor that kept up with the changes in demand. The lack of wage premiums may also be explained by other factors, including the role that technology has played in automating various dispensing and administrative roles that had previously been performed by pharmacy technicians, as well as negative pressures on pharmacy revenues generated by dispensing, which are needed to pay employee wages and other expenses.

## 5. Conclusions

The period from 1997 to the present has been one of transition for pharmacy across all settings in the United States, with increased demands for patient safety, access to safe and effective medication therapy, and value for what payers obtain from medications. Pharmacy technicians have been an essential component of these transitions, with pharmacies and pharmacists increasingly depending on them to support the delivery of high-quality patient care. This study provides evidence that compensation levels for the majority of pharmacy technicians in the United States have not increased in line with changes in the US consumer price index, nor have they increased in line with their increased responsibilities. It is important that the compensation, and particularly the hourly wage levels, of pharmacy technicians continue to be evaluated. This evaluation is essential to maintaining and supporting this important segment of the pharmacy workforce.

## Figures and Tables

**Figure 1 pharmacy-08-00042-f001:**
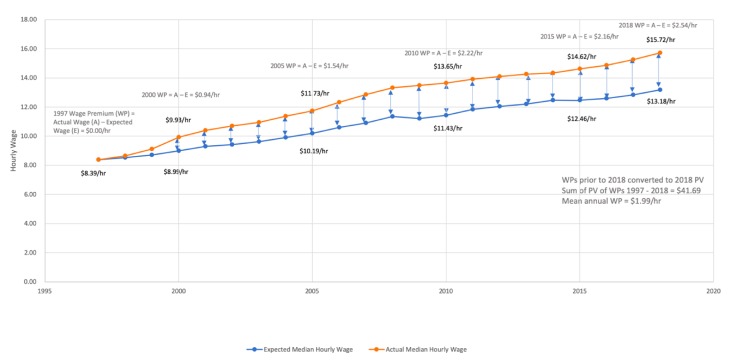
Calculation of the Mean Annual Hourly Wage Premium for Pharmacy Technicians from Base-year 1997 to 2018.

**Figure 2 pharmacy-08-00042-f002:**
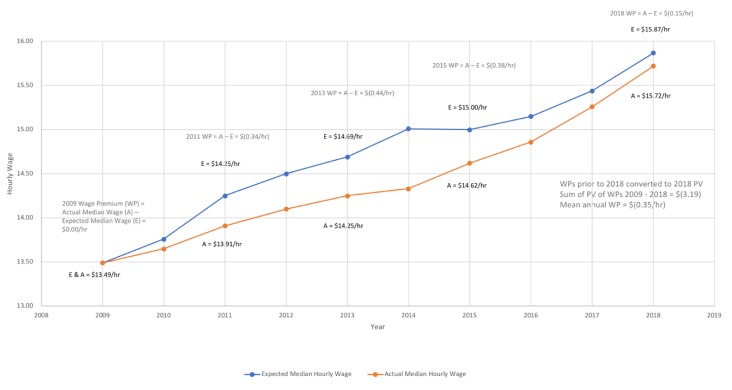
Calculation of the Mean Annual Hourly Wage Premium for Pharmacy Technicians from Base-year 2009 to 2018.

**Figure 3 pharmacy-08-00042-f003:**
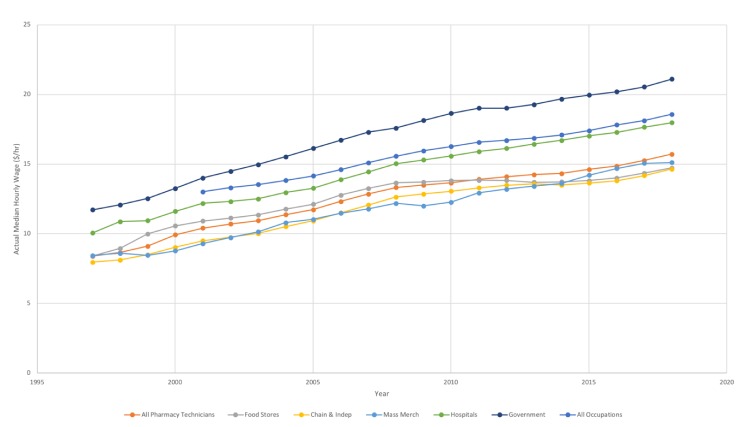
Median Hourly Wages for Pharmacy Technicians, by work setting, from 1997 to 2018.

**Figure 4 pharmacy-08-00042-f004:**
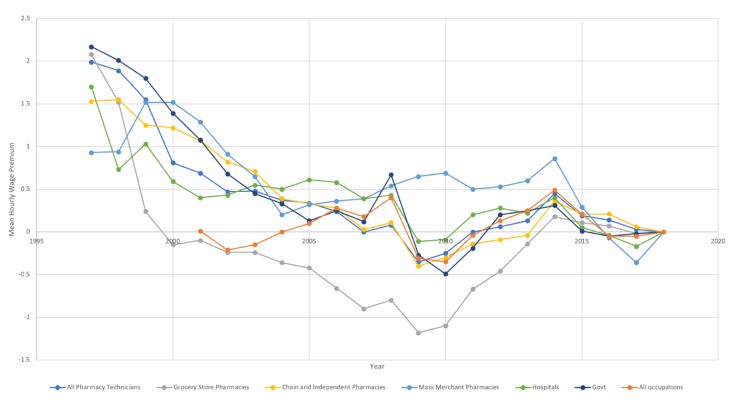
Mean Annual Hourly Wage Premiums for Pharmacy Technicians, from 1997 to 2018, and All Occupations, from 2001 to 2018.

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
