# Peer review of "Wage Premiums as a Means to Evaluate the Labor Market for Pharmacy Technicians in the United States: 1997–2018"

_pharmacy, 2020, doi:10.3390/pharmacy8010042_

Round 1

Reviewer 1 Report

Overall, I think this is a very interesting and important topic.

Methods

  • There's some information in the methods section that seems more like results - Figures 1 and 2
  • In Figure 1 the expected median hourly wage is reported as $12.46 and the actual median hourly wage is $14.62.  In Figure 2 the actual median hourly wage stayed the same but the expected median hourly wage is now reported as $15.  It is unclear how this change occurred.  

Figure 1 and Figure 2

  • A little hard to read, the text seems to get a bit lost with the grid lines.
  • It can be assumed what each abbreviation is representing but just for clarity would recommend adding a footnote to identify the abbreviations.

Discussion

  • I think the discussion needs to be built out a bit to highlight why this matters.  For example, it would be useful to discuss the trend that wage premiums have been decreasing.  The authors state in the methods that the wage premium should be $0 so couldn't the argument be made that positive wage premiums are not a good thing since this implies that technicians are being overpaid or there is a shortage of technicians so salaries are higher?
  • While this may be outside the scope of this paper, I think a discussion about how the role of the technician has changed since 1997 but the salaries are still remaining relatively consistent with increases likely attributable to CPI.  One of the points I found to be interesting was the comparison of salaries in 1997 to 2018 using 2018 dollars.

Reviewer 2 Report

Thank you for the submission.  The work is quite interesting. However, there are some holes in the logic, which is essentially suggesting that the pharmacy technicians are underpaid and/or their wages have not increased at an adequate rate (given their expanded roles). Yet, Figure 4 doesn’t show this. And we are not told anything about the actual distribution of roles in the workforce (especially in community pharmacies). Do we know the distribution of roles across the population of pharmacy technicians? What percentage actually performs clinical roles such as “taking medication histories, managing warfarin therapy within a clinical pharmacy anticoagulation service and immunization delivery”. Has the role become more complex, on average? This might be reflected in the temporal trend in the percentage of pharmacy technicians performing these more complex activities. But we don’t have these data presented. If the vast majority of technicians are performing relatively simple functions (especially in community pharmacy), should we expect the wages to have increased substantially?  

How have the wages changed in relation to US pharmacist wages over the same period?

All this is needed to provide some important context, especially for international readers. I come from a country where community pharmacist wages have declined relative to CPI over the past 20 years, so I don’t know what to make of these types of statements: “Given the median length of employment of pharmacy technicians, it is likely that the majority of technicians working in US pharmacies have not experienced increases in their wages relative to what would be expected by changes in the CPI. This has occurred at a time when pharmacies and pharmacists are asking more of their pharmacy technicians.”

In fact, for most of the manuscript the words “pharmacy technicians” could be replaced by “pharmacists” and the same would apply, at least in the countries I am most familiar with (e.g. “Mattingly and Mattingly also concluded in their systematic review of literature regarding the roles of pharmacy technicians that while evidence supports technicians performing roles which advance pharmacy practice and improve patient outcomes, the benefits to technicians in performing these roles have been limited to increases in their job satisfaction and work schedules, and not in their levels of wages or other forms of financial compensation.") I guess I am looking for greater clarity around whether the situation is any different for pharmacists?  There is only a brief mention of this late in the manuscript: “Pharmacy technicians by and large have not experienced wage changes in the manner of those experienced by pharmacists, who experienced relatively large wage premiums in the late 1990s through the late 2000s, and have since seen their wages change in line with changes in the CPI [28].” This should be shown in Figure 4.

A more useful objective for the manuscript would be: The objective of this research is to use BLS data to evaluate changes in pharmacy technician wages from 1997 to 2018 relative to changes in the US Consumer Price Index (CPI) and pharmacist wages over that time.

The analysis and associated number of figures is overly complicated and long. Expressing the wages as a “premium” relative to CPI is simply an alternative (and inefficient) way of expressing the same thing. It does not need a doubling of the Figures to tell readers how wages have tracked relative to CPI. There is no additional value in the analysis in Figure 1, and it should be deleted.

Figure 5 should also be deleted; the necessary information is already in the text and in Figure 3.

The manuscript needs minor changes throughout to reflect that it is a USA-only study. e.g. “The labor market for pharmacists in the US has been evaluated by means of the Aggregate Demand Index (ADI) [16], and later by the Pharmacist Demand Indicator (PDI) [17]. Pharmacist demographics, working conditions and other trends in their practice are examined in the US National Pharmacist Workforce Survey…”

Round 2

Reviewer 2 Report

Thank you for the revision.